# STOCHASTIC SAFE ACTION MODEL LEARNING

## ABSTRACT

Hand-crafting models of interactive domains is challenging, especially when the dynamics of the domain are stochastic. Therefore, it's useful to be able to automatically learn such models instead. In this work, we propose an algorithm to learn stochastic planning models where the distribution over the sets of effects for each action has a small support, but the sets may set values to an arbitrary number of state attributes (a.k.a. fluents). This class captures the benchmark domains used in stochastic planning, in contrast to the prior work that assumed independence of the effects on individual fluents. Our algorithm has polynomial time and sample complexity when the size of the support is bounded by a constant. Importantly, our learning is safe in that we learn offline from example trajectories and we guarantee that actions are only permitted in states where our model of the dynamics is guaranteed to be accurate. Moreover, we guarantee approximate completeness of the model, in the sense that if the examples are achieving goals from some distribution, then with high probability there will exist plans in our learned model that achieve goals from the same distribution.

## 1 INTRODUCTION

In classical (high-level) task planning problems, a domain model describes the the interaction between an environment and the planning agent. A domain model is usually specified in a formal language and includes an action model, which specifies which actions can be in a plan and how they work. Such formal languages include STRIPS (Fikes & Nilsson, 1971) and PPDDL(Aeronautiques et al., 1998), for example. The action model describes the effects of the actions on the environment's state, and the preconditions that must be true in order for the action to be taken. Creating a planning domain model and action model, however, is a notoriously hard knowledge-engineering task. To overcome the modeling problem, many approaches have been proposed to automatically learn the domain model (Yang et al., 2007; Cresswell & Gregory, 2011; Zhuo & Kambhampati, 2013; Stern & Juba, 2017; Aineto et al., 2019; Juba et al., 2021; Juba & Stern, 2022).

This problem is even more difficult when the domain dynamics is stochastic (Juba & Stern, 2022). Indeed, in a deterministic environment, we can learn that some state attributes are not part of the effect as long as they are not in the post transition state, and that they are part of an effect as long as it appears in the post state but not the previous state. However, when the effects are stochastic, it's possible that an effect does not appear in the post state and yet has significant probability of appearing. Consequently, we will need many observations; if there are even small errors in our estimates of these probabilities, it can accumulate over the course of execution, leading to wildly inaccurate estimates of the trajectory. Moreover, even in these simple domain models where effects simply set some fluents to take specific values, we may not observe whether or not an effect occurred if the fluent already had that value prior to taking the action. Therefore, in any given transition, we generally cannot know for certain which fluents would have been set by the random effects of the action, even when the values of the fluents are fully observed.

Our goal is to safely learn an stochastic action model that is guaranteed to be accurate and complete. For safety, we use *offline learning* of the action model, assuming that demonstrations of competent performance of the domain have been provided e.g., by a human controller. We also seek a guarantee that the model is sufficiently accurate to capture and avoid potential danger during planning. This is similar to offline reinforcement learning (Levine et al., 2020), except that we wish to learn a reusable model that allows solving many goals, and we learn high-level task planning representations (with concise PPDDL representations). To ensure accuracy, we produce a conservative model that

only permits actions to be taken when the learned model can be guaranteed to accurately describe the actual distribution of possible transitions in the environment. We must further ensure that this conservative model is permissive enough to allow successful performance in the domain. Previous work that tackles safe learning of high-level task planning models of stochastic environments (Juba & Stern, 2022) achieved safety and completeness under the assumption that the effects of actions on each fluent are independent random variables. In this work, we relax that assumption and provide the algorithm for a general stochastic environment.

We note that the prior work by Juba & Stern (2022) gave an efficient and safe method for learning the preconditions of the actions that carries over to this more general problem. So, we focus on the problem of learning the distribution of the random effects for each action $a$. Specifically we will use method of moments originally pioneered by Pearson (1936) (Wooldridge, 2001; Hall, 2004; Ney, 1985; Gibson, 2021; Newey & West, 1987; Ogaki, 1993; Mátyás et al., 1999, for example) to learn the distribution of effects. In particular, work by Anandkumar et al. (2014) demonstrated that the method of moments may be efficiently realized using algorithms for tensor decomposition to fit a wide class of probabilistic models, given a certain "identifiability" assumption: that the moments uniquely determine the parameters. When the parameters are "generic" numbers, identifiability is guaranteed; unfortunately, no number with a finite representation is "generic." Worse, in our setting, the full effects distribution actually may not be identifiable.

In this work, we overcome these challenges as follows: first, we observe that for the kinds of simple discrete effects used in classical planning models (e.g., as appearing in the International Planning Competition probabilistic planning track), a moderate number of moments suffices to ensure identifiability of fully observed distributions (Sec. 2.3). Second, we observe that for an accurate model of the domain, since the "unobserved" effects are those that leave the fluents with the same values, it is sufficient for the observed marginals to be consistent. Third, we give a polynomial-time algorithm to construct a set of stochastic effects that is consistent with the observed marginals. Finally, we show that we can add some additional preconditions that ensure that the domain model only permits actions to be taken when the effect distribution in our learned model is guaranteed to be close to the true distribution; in particular, we argue that this does not significantly reduce the completeness of the model, relative to the distribution on example trajectories.

## 1.1 RELATED WORK

A few approaches to learning planning models for stochastic domains have been previously proposed. Pasula et al. (2007) formulated their learning problem as optimizing an objective that could not be tractably optimized, and proposed a greedy heuristic solution. Unfortunately, this heuristic cannot provide the soundness or approximate completeness guarantees that we seek. On the other hand, Mourão et al. (2012) assumes that the actual domain is deterministic, and merely the observations are corrupted by stochastic noise. (They also do not provide the kind of guarantees that we seek.) More recently, Mao et al. (2022) proposed to use a neural network model to predict missing parts of an action model. While presumably quite powerful, this again comes at the price of any expectation of soundness and prevents the representation from being used in standard planners.

Our algorithm for constructing consistent set of effects solves a problem resembling the analogue of low-rank matrix completion (e.g., Candès & Recht (2012)) for tensors with a nonuniform observation model. Prima facie, this is not possible since the solution may not be identifiable. As we discuss above, it is crucial for our problem that it is sufficient to only match the distributions on marginals for which we have observations. The "completed" entries may, in general, be wildly off, and so the additional preconditions are necessary to ensure that the model only permits policies to take actions where the distribution of their effects is guaranteed to be modeled correctly. The upshot is that it would not be possible for us to use an "off-the-shelf" algorithm for low-rank tensor completion for our problem such as Yang et al. (2021), because the lack of identifiability in general ensures that at least some of our instances would not satisfy the assumptions required for those algorithms.

## 2 PRELIMINARIES

We now recall our problem domain and the Stochastic Safe Action Model Learning (SAM) problem. Subsequently, we will introduce some mathematical tools for the method of moments.

## 2.1 STOCHASTIC SAFE ACTION MODEL LEARNING PROBLEM

We formulate our problem in terms of grounded PPDDL representations for simplicity. We also do not consider conditional effects or action costs. A domain $D = \langle F, A, M \rangle$ consists of a set $F$ of *fluents*, a set $A$ of *actions*, and an *action model* $M$ for these actions. A fluent $f$ is a variable representing a fact that may or may not hold in the environment at some point in time. A state $s$ is a vector of assignments of Boolean values $s(f)$ to each respective fluent $f$ in $F$. We will abuse notation by denoting the indicator functions $1[s(f) = 1]$ as $f$ and $1[s(f) = 0]$ as $\neg f$, which we refer to as *literals*.

An action is an operation that can be taken by the planning agent to change the values in $s$, hence transitioning to another state $s'$. The action model $M$ defines *preconditions* $pre_M(a)$ and *effects* $eff_M(a)$ for each action $a \in A$. A precondition of an action is a Boolean formula on the fluents, with the interpretation that the precondition must be satisfied on the current state $s$ to allow the action $a$ to be taken. In the domains we consider, these preconditions will be conjunctions, i.e., an AND of literals, but we will produce action models in which the preconditions are conjunctive normal form formulas: an AND of ORs of literals.

We consider a fragment of PPDDL in which the effects have a single "`probabilistic`" block for each action: this means that for each action, there is a sequence of $r$ partial assignments $e_1, e_2, \ldots, e_r$ where each $e_i$ is associated with a respective probability $p_i$. That is, when action $a$ is taken, with probability $p_i$, the partial assignment $e_i$ will occur in the next state $s'$, and any fluents not set in $e_i$ remain the same value as in the previous state $s$. We assume that the number of effects $r = r(a)$ for each $a$ is at most a small constant, which is consistent with the benchmarks used in the IPC probabilistic tracks, where the number of effects is generally below five. Note that the action model thus specifies the probability of transitioning from a state $s$ to another state $s'$ by applying $a$, denoted by $Pr_M[s'|a, s]$.

A PPDDL planning problem $\Pi = \langle D, s_I, s_G \rangle$ consists of a domain $D$, a starting state $s_I$, and a goal state $s_G$. A solution to a PPDDL planning problem is a policy $\pi$, which is a mapping from the states to the actions $\pi : 2^F \to A$. To execute a policy $\pi$ on the planning problem $\Pi$ is to repeatedly apply actions according to the policy $\pi$ given the current state, starting by applying action $\pi(s_I)$ at the starting state $s_I$. The execution ends when the goal state $s_G$ is reached, or some other condition is satisfied, such as a time-out number of actions taken, after which the agent gives up. A trajectory $\mathcal{T}$ is a an alternating sequence of states and actions of the form $\langle s_0, a_0, s_1, a_1, \ldots, a_{|\mathcal{T}|}, s_{|\mathcal{T}|} \rangle$. We assume the trajectory includes the values of each fluent at each state, and an identifier of which action was taken for each transition. Each execution of a policy $\pi$ creates a trajectory starting from $s_0 = s_I$. The length $|\mathcal{T}|$ of trajectory $\mathcal{T}$ is the number of actions taken.

In the Stochastic Safe Action Model (SAM) Learning problem, we suppose that there is an arbitrary probability distribution on problems in a fixed domain $D$ and policies for $D$. Trajectories $\mathcal{T} = \{\mathcal{T}_1, \ldots, \mathcal{T}_m\}$ are sampled by first drawing an independent $i$th pair of problem $\langle D, s_I^{(i)}, s_G^{(i)} \rangle$ and policy $\pi^{(i)}$ from the distribution, and then executing $\pi^{(i)}$ in $D$ from $s_I^{(i)}$ to produce $\mathcal{T}_i$. We are given this sample of trajectories (implicitly with identifiers for the names of fluents $F$ and actions $A$) as input. We produce as output an action model $\hat{M}$, thus giving a learned domain representation $\hat{D} = \langle F, A, \hat{M} \rangle$. For a given $\epsilon$ and $\delta$, we require that with probability $1 - \delta$, we obtain $\hat{M}$ such that

1. *"Safety"* For any policy $\pi$ that takes $L$ legal actions in $\hat{D}$ in expectation, the distribution on trajectories obtained by $\pi$ in $\hat{D}$ is $\epsilon$-close in total variation distance to the distribution obtained by $\pi$ in $D$, and the actions of $\pi$ are legal in $D$ as well.

2. *"Approximate completeness"* For problems $\langle D, s_I', s_G' \rangle$ and policies $\pi'$ sampled independently from the training distribution, with probability $1 - \epsilon$ there is a policy $\hat{\pi}$ that takes legal actions in $\hat{D}$ and such that the probability that $\hat{\pi}$ reaches $s_G'$ from $s_I'$ in $\hat{D}$ is less than the probability that $\pi'$ reaches $s_G'$ from $s_I'$ in $D$ by at most $\epsilon$.

In our guarantees, in addition to the running time of an algorithm for this problem, we must show that there is a polynomial bound on the number of trajectories $m = m(|F|, |A|, L, \epsilon, \delta)$ needed to obtain such an action model with probability $1 - \delta$. This is the *sample complexity* of the problem.

## 2.2 Tensor decomposition

A degree-$d$, dimension-$n$ tensor $T$ is an array of numbers $T[i_1, \ldots, i_d]$ indexed by $d$ indices, where each index $i_j \in [n]$ for $j \in [d]$. For example, a matrix is a degree-2 tensor. To decompose a tensor $T$ is to represent each of its element as the weighted sum of $r$ products:

$$T[i_1, \ldots, i_d] = \sum_{k=1}^{r} w_k v_k^{(1)}[i_1] \cdots v_k^{(d)}[i_d],$$

for vectors $v_k^{(j)}$ of dimension $n$. Equivalently, we can write it as: $T = \sum_{k=1}^{r} w_k v_k^{(1)} \otimes \cdots \otimes v_k^{(d)}$, where $\otimes$ is the outer product of two vectors. The minimum number of terms $r$ in such possible decompositions is called the tensor rank of $T$. We refer to the set of vectors $V^{(j)} = [v_1^{(j)}, \ldots, v_r^{(j)}]$ as the $j$th mode of this tensor decomposition, where $j \in [d]$. If all modes $V^{(j)}$ are the same $V = [v_1, \ldots, v_r]$, we can write the decomposition as

$$T = \sum_{k=1}^{r} w_k v_k^{\otimes d}, \tag{1}$$

where $v_k^{\otimes d}$ is the outer product of a vector $v_k$ with itself $d$ times. Note that for positive $w_k$ (or odd $d$), by rescaling each vector $v_k$ by $w_k^{1/d}$, we can obtain the same tensor while dropping $w_k$.

We know that in general, matrix decompositions are not unique. By contrast, the decomposition for any "generic" tensor is known to be unique. This is a useful property, since it enables reconstructing the components of a mixture of distributions in the method of moments, for example. However, such guarantees for "generic" tensors have an exception of a set of tensors with measure zero. Since the tensor we want to decompose involves binary vectors, which are discrete, they are not "generic" and hence we cannot use such guarantees for "generic" tensors. In order to establish that the tensor decomposition is unique, we leverage Kruskal's theorem Kruskal (1977). It is the cornerstone of establishing identifiability in many settings (Gu, 2022; Allman & Rhodes, 2008; Fang et al., 2019; Culpepper, 2019; Chen et al., 2020; Fang et al., 2021; Xu, 2017).

**Theorem** (Kruskal (1977)). *Suppose that a degree-3 tensor $T$ has a decomposition $\sum_{k=1}^{r} a_k \otimes b_k \otimes c_k$. Let $A = [a_1, \ldots, a_r]$, $B = [b_1, \ldots, b_r]$, and $C = [c_1, \ldots, c_r]$ denote matrices with these vectors as columns. Suppose every set of $I$ columns of $A$ are linearly independent, every set of $J$ columns of $B$ are linearly independent, and every set of $K$ columns of $C$ are linearly independent. If $I + J + K \geq 2r + 2$, then this tensor decomposition involving $r$ components is unique up to permutation.*

## 2.3 Unique Tensor Decomposition for Boolean Components

We now show that for any tensor power of Boolean components that is logarithmic in the number of components, the tensor decomposition is unique. Thus, for the tensors corresponding to logarithmic moments of a discrete distribution, the tensor decomposition recovers the components. This can be implicitly seen in the results of Chen & Moitra (2019), but for completeness we will present it here and include a full proof in the appendix. Suppose that we are given a degree-$2k + 1$ tensor $\sum_{i=1}^{r} w_i v_i^{\otimes(2k+1)}$, where $v_i \in \{0, 1\}^d$. The goal is to obtain $v_i$. We will re-shape it and obtain a degree-3 tensor $\sum_i^r w_i v_i \otimes flat(v_i^{\otimes k}) \otimes flat(v_i^{\otimes k})$. Here $flat(M)$ means that we rearrange the tensor into a vector.

By Kruskal's Theorem, we can argue that the tensor decomposition of $\sum_i^r v_i \otimes flat(v_i^{\otimes k}) \otimes flat(v_i^{\otimes k})$ is unique up to permutation. Indeed, suppose $V = \{v_1, \ldots, v_r\}$, and $V^{\otimes k} = \{flat(v_1^{\otimes k}), \ldots, flat(v_r^{\otimes k})\}$. According to Lemma 1, $V^{\otimes k}$ must be full rank when $k = O(\log(r))$:

**Lemma 1.** *For all $k, n \in \mathbb{N}$ and $S \subseteq \{0, 1\}^n$, if $|S| \leq 2^{k+1} - 2$, then $S^{\otimes k}$ is linearly independent.*

So, if $V$ has at least two distinct 0-1 vectors, we can guarantee

$$rank(V) + rank(V^{\otimes k}) + rank(V^{\otimes k}) \geq 2 + 2r.$$

Due to Kruskal's Theorem, when $k = O(\log(r))$, the decomposition is unique. Although the statement of Kruskal's Theorem does not include weights $w_i$, we can recover these from the scaling:

Since $v_i$ is known to be a 0-1 vector, we can still read out the 0-1 information from the zero and nonzero values of the decomposed vectors, and the value of $w_i$ will be the product of the the nonzero values of the three modes for each $i$. Hence, we can obtain our decomposition by computing the tensor decomposition of this $d \times kd \times kd$ reshaping using any tensor decomposition method.

## 3 APPROACH TO STOCHASTIC SAM LEARNING

We now give an overview of our approach to solving the Stochastic SAM Learning problem.

### 3.1 LEARNING PRECONDITIONS

Preconditions can be learned following the same approach described in Juba & Stern (2022): For the class of domains we consider, for each $(s, a, s') \in T$, where $T \in \mathcal{T}$, we have that if a literal is not in the previous state then it cannot be a precondition. Precisely: $\forall \ell : s(\neg \ell) \Rightarrow \ell \notin pre(a)$ where $pre(a)$ is the preconditions of action $a$ according to the actual action model $M^*$. We can thus learn the preconditions by initially assuming every action has all the literals as preconditions, and then applying this rule to remove literals from the preconditions as needed. More specifically, let $\mathcal{T}(a)$ be all the $\langle s, a, s' \rangle$ triplets for action $a$. States $s$ is the pre-state and $s'$ is the post-state of action $a$. We (initially) set the preconditions of $a$ to be $pre(a) = \{\ell : \forall \langle s, a, s' \rangle \in \mathcal{T}(a) \ s(\ell)\}$.

### 3.2 LEARNING EFFECTS

We now need to recover the set of effects $\{e_i\}$ for each action $a$, where each effect $e_i$ has a corresponding probability $p_i$. These estimated stochastic effects, together with our learned preconditions, will give an approximate action model $\hat{M}$ solving the Stochastic SAM Learning problem. To recover the effects from our observations, we first estimate moments of the effect indicator variables, i.e., where the indicator for literal $\ell$ is 1 if $\ell$ was an effect of $a$ and 0 otherwise. The values of these indicators can only be determined from the data if $\ell$ was not true in the previous state. Since the distribution of effects is assumed to only depend on the action taken (and not the previous state), we can estimate these probabilities by conditioning on $\ell$ being false in the previous state, i.e., counting the number of such transitions in the example trajectories. The problem that will arise is that we may not have such states where $a$ is taken and $\ell$ is false; we will return to this issue later.

For the purpose of illustration, let's start with degree-3 moments: Let $n = 4$ be the total number of literals $\ell$. Suppose action $a$ has $r = 3$ effects: $e_1 = \{\ell_1, \ell_3\}$, $e_2 = \{\ell_1, \ell_2\}$, and $e_3 = \{\ell_2\}$. the probability values we observed and approximated are

$$p_{\ell_i, \ell_j, \ell_k} := Pr\left[s'(\ell_i), s'(\ell_j), s'(\ell_k) | s(\neg \ell_i), s(\neg \ell_j), s(\neg \ell_k), a\right]. \tag{2}$$

These degree-3 moment data can be arranged into a 3-dimensional tensor $T_a$ where each entry is the probability $p_{\ell_i, \ell_j, \ell_k}$ as in Equation 2 and indexed by three literals $\ell_i, \ell_j, \ell_k$. Its value will be the *sum of all $p_i$ for all effects $e_i$ that induces $\ell_i, \ell_j, \ell_k$ being true* under the condition that $\neg \ell_i, \neg \ell_j, \neg \ell_k$ are true. We can then extract the effects from these moment data. Indeed, the effects we want to recover can be formulated as 0-1 vectors:

$$e_1 = (1, 0, 1, 0), \ \ e_2 = (1, 1, 0, 0), \ \ e_3 = (0, 1, 0, 0)$$

The degree-3 moments can arranged into a degree-3 tensor as follows:

$$T_a = \sum_{i=1}^{r} p_i e_i^{\otimes 3} = p_1 e_1^{\otimes 3} + p_2 e_2^{\otimes 3}, + p_3 e_3^{\otimes 3}, \tag{3}$$

where $p_i$ are the probabilities of these effects. Since Eq. 3 is a feasible decomposition of $T_a$, when it is unique, it must be the only decomposition $T_a$. Thus, if we compute a tensor decomposition of $T_a$, the vectors $e_1$, $e_2$, and $e_3$ in the decomposition are the effects of the action $a$, with the associated scalings $p_1, p_2, p_3$ as the probabilities of the respective effects. As long as the number of effects $e_i$ is small and the components are linearly independent (established in Sec. 2.3), then this problem can be solved in polynomial time by existing algorithms (for example, see Schramm & Steurer (2017) or Anandkumar et al. (2014)). Since the number of entries in this tensor is $O(n^3)$, we can find this decomposition in $O(poly(n))$ time.

We find that the decomposition is indeed unique, following Sec. 2.3. Depending on the number of effects $r$ we have, we construct moments of degree $d = O(\log r)$:

$$p_{\ell_{i_1}, \ldots, \ell_{i_d}} := Pr[s'(\ell_{i_1}), \ldots, s'(\ell_{i_d}) | s(\neg \ell_{i_1}), \ldots, s(\neg \ell_{i_d}), a] \qquad (4)$$

and the tensor decomposition we want to obtain is the following:

$$T_a = \sum_{i=1}^{r} p_i e_i^{\otimes d}, \text{ where } e_i \in \{0, 1\}^n, \forall i \in [r]. \qquad (5)$$

For such $d$, following Sec. 2.3, we can reshape this tensor into a degree-3 tensor and use the existing tensor decomposition algorithms to obtain the unique decomposition solution vectors. Importantly, since this decomposition is unique, and the 0-1 vectors $e_i$ are a feasible solution, it must be the only solution. Hence we can retrieve the effect vectors for $a$ from $T_a$. The size of $T_a$ is $n^{O(\log(r))}$, so our tensor decomposition algorithm will take $poly(n^{O(\log(r))})$ time to run. As long as $r$ is a small constant (not scaling with $n$), this may still be computed in polynomial time.

To empirically estimate each entry given by Eq. 4 in our moment tensor, we count the number of times it occurs. This will give us a $1 - \delta$ confidence interval that contains the correct moment value (w.p. $< \delta$, it deviates from the true value by $> \epsilon$):

$$p_{\ell_{i_1}, \ldots, \ell_{i_d}} = \frac{\#\langle s, a, s' \rangle \in \mathcal{T}(a) : s'(\ell_{i_1}), \ldots, s'(\ell_{i_d}), s(\neg \ell_{i_1}), \ldots, s(\neg \ell_{i_d})}{\#\langle s, a, s' \rangle \in \mathcal{T}(a) : s(\neg \ell_{i_1}), \ldots, s(\neg \ell_{i_d})} \pm \Delta(\epsilon, \delta), \qquad (6)$$

where $\Delta(\epsilon, \delta)$ is a small positive number depending on $\epsilon$ and $\delta$. We will take the empirical count (mid-point of the interval) as our entry for the estimated tensor $T_a$ (see appendix for a discussion).

The caveat of this is that our tensor may have many missing entries. For the entry that corresponds to $\ell_{i_1}, \ldots, \ell_{i_d}$, we count the appearing frequency when all three literals are affected by this action. That is, we need to observe $s(\neg \ell_{i_1}), \ldots, s(\neg \ell_{i_d}) = 1$, and $s(\ell_{i_1}), \ldots, s(\ell_{i_d}) = 0$ in the previous state $s$ changing into $s'(\ell_{i_1}), \ldots, s'(\ell_{i_d}) = 1$, and $s'(\neg \ell_{i_1}), \ldots, s'(\neg \ell_{i_d}) = 0$ in the next state $s'$, due to the action $a$ being taken. If such a pre-state never appears (or does not appear enough times), then we do not have a valid estimate, and this entry will be considered missing.

Observe that when a literal $\ell$ is true in the pre-state, it does not matter whether $\ell$ is an effect of the action; as long as $\neg \ell$ is *not* an effect, $\ell$ will remain true. Thus, for any given state $s$, if we consider the minor of the tensor $T_a$ given by the indices of literals that are false in $s$, it is enough for this minor to be fully observed: the decomposition of the corresponding (minor of the) tensor identifies the distribution of post-states for the action $a$. Note that these minors of the tensor form blocks that are symmetrical w.r.t. the literals used as indices. For example, if the tensor is a matrix, then the effect set $\{\ell_i, \ell_j\}$ is represented in the identical entries $(\ell_i, \ell_j)$ and $(\ell_j, \ell_i)$. Therefore, to obtain an accurate action model, it is sufficient to find a distribution of effects that is consistent with these minors of the tensor. In the sections below we will discuss how to decompose the tensor with many missing entries.

To guarantee safety, we add additional $d$-CNF preconditions that ensure that the agent only uses action $a$ for these fully-observed minors: for each missing entry of the tensor $T_a$ corresponding to the literals $\ell_{i_1}, \ldots, \ell_{i_d}$, we add a clause $\ell_{i_1} \vee \ldots \vee \ell_{i_d}$ to $pre(a)$. Note that by De Morgan's law, this is equivalent to the condition that not all of $\ell_{i_1}, \ldots, \ell_{i_d}$ are 0, so for any state $s$ satisfying the precondition, the minor for the literals that are false in $s$ does not include this missing entry. Since entries are only considered missing when states for which all of $\ell_{i_1}, \ldots, \ell_{i_d}$ are 0 rarely occur in the training set, we will be able to guarantee that this additional precondition does not significantly reduce the completeness of the action model.

## 4 Algorithm for Learning Stochastic Effects

Since there are missing entries in the data tensor $T_a$ we construct for each action, we can not directly apply existing tensor decomposition algorithms to find a tensor decomposition to recover the effect vectors. Nevertheless, we saw that we only need to produce a distribution of effects that is consistent with the moments that we can estimate. In this section, we will use this observation to develop an algorithm that first decomposes observed minors of the tensor, then combines these partial effect vectors to obtain a global effect vector.

Our algorithm is Alg. 1. We will give a detailed discussion for each component of the algorithm in the following subsections.

---

**input :** $O(\log(r))$-degree moments where each entry is the empirical estimate of Eq. 4, states $s \in \mathcal{T}(a)$.
**output:** global effect vectors $\{e\}$ and their probabilities $\{p\}$

1 **begin**
2      Reshape the moment tensor into degree-3 tensor.
3      Draw a random Gaussian vector $g$ and contract the tensor blocks $B_s$ for each $s \in \mathcal{T}(a)$ to matrices using $g$
4      Compute the tensor decomposition for each block and obtain the 0-1 local effect vectors $\hat{e}_i$
5      **while** *not all blocks have been reduced to zero* **do**
6          **while** *not all blocks have been tightened and their constraints dropped* **do**
7              Fix a new $\lambda$: geometric search for its upper bound, then binary search for tightness.
8              Solve the SDP in Eq. 7,
9              Find the tight blocks $B_s$, and obtain their eigenvectors $x_i$.
10              Un-whiten each $x_i$ to obtain $e_i$, and eliminate all $\hat{e}_{i'}$ that are inconsistent with $e_i$ by using Eq. 8. Drop the tight $B_s$s' constraints.
11          **end**
12          Combine un-whitened $e_i$ vectors into a global effect vector $e$.
13          Collect $p = \lambda_m / \langle g, e_{i_m} \rangle$ as the probability for the current global effect, where $\lambda_m = \min_i \lambda_i$.
14          Subtract $\lambda_m e_i^{\otimes 2}$ from their corresponding blocks.
15      **end**
16 **end**

**Algorithm 1:** Stochastic Effect Learning

---

We leverage the fact that these non-missing minors are symmetric to perform minor-wise decomposition, where the local solution of a minor is always consistent with a global solution $v_i$. Hence we can formulate constraints to make the these local pieces consistent with a solution $v_i$ when they are pieced together. Because the full tensor is low rank, the minor must also be low rank. We can use existing methods to decompose them individually.

In order to efficiently piece them together, we will iteratively pull out the top eigenvector from each block, and eliminate the vectors in other blocks that are inconsistent with it. Next round we will only pull out a top eigenvector that is consistent with the previous vectors. Combining them together, we obtain a global effect vector.

## 4.1 LOCAL DECOMPOSITION ALGORITHM

As long as we have uniqueness, we can use any suitable tensor decomposition method to obtain local 0-1 effect vectors from the minors. We will recall Jennrich's algorithm (Harshman, 1970; Leurgans et al., 1993) as the starting point for our combining method. We first reshape the higher degree tensor into degree-3 tensor. So, suppose we are given a tensor $T \in \mathbb{R}^{n^3}$, and it has decomposition $T = \sum_{i=1}^{r} a_i^{\otimes 3}$ with orthogonal vectors $a_i, \ldots, a_r \in \mathbb{R}^n$. Then we can compute its this decomposition in the following steps:

Firstly, pick a Gaussian random vector $g \in \mathbb{R}^n$, and compute the projection of $T$ onto $g$:

$$T_u = \sum_{j=1}^{n} g_j T[j, :, :] = \sum_{j=1}^{n} \sum_{i=1}^{r} (g_j \cdot a_{i,j}) \cdot a_i \otimes a_i = \sum_{i=1}^{r} \langle g, a_i \rangle \cdot a_i \otimes a_i = \sum_{i=1}^{r} \langle g, a_i \rangle \cdot a_i a_i^\top.$$

Here $T[j, :, :]$ means the $j$th slice of tensor $T$. It can be viewed as a weighted sum of all the slices of matrices of the tensor $T$. This process is also called contraction.

Then, we compute the singular value decomposition of $T_g$. Note that since $a_1, \ldots, a_r$ are orthogonal, they are a set of candidate eigenvectors of $T_g$. Moreover, since $g$ is Gaussian, the values $\langle g, a_i \rangle$ are going to be distinct with probability one, and hence this singular value decomposition is unique. The computed eigenvectors must be $a_1, \ldots, a_r$.

In our case, $a_i = e_i$. (Note that Jennrich's algorithm requires the rank $r \leq n$, as we assume here.)

However, this requires $a_1, \ldots, a_r$ to be orthogonal. In our case $a_i = e_i$. Following Sec. 2.3, the 0-1 effect vectors are linearly independent when the degree of the moments is high enough, so we can pre-process the tensor by whitening. That is, we will apply a whitening matrix $W$ and compute:

$$T_g^W = W T_g W^\top = \sum_{i=1}^r \langle g, e_i \rangle \cdot W e_i e_i^\top W^\top = \sum_{i=1}^r \langle g, e_i \rangle \cdot (W e_i)(W e_i)^\top.$$

Here, with the abuse of notation, we use $e_i$ to also denote itself raised to tensor power $d = O(\log(r))$, $flat(e^{\otimes d})$, when the context is not confusing. Therefore, for the degree-2 moment matrix $M = \sum_{i=1}^r e_i e_i^\top$, we choose $W = M^{-1/2}$. One method of computing $W$ is through PCA. So we will be computing vectors $a_i = W e_i$ by tensor decomposition. Then we can retrieve $e_i$ by computing $e_i = W^{-1} a_i$. We will refer to this as un-whitening in the following discussion.

## 4.2 COMPOSING THE FRAGMENTS

We will first sample a random Gaussian vector $g$ to contract all the blocks, and pre-compute the tensor decomposition for each block and obtain their unique local tensor decomposition vectors $\{\hat{e}_i\}$. To obtain the global effect vector, we will extract one global eigenvector at a time, then subtract it from the blocks by a common weight, similar to eigendecomposition. Since what we have is a set of contracted blocks $\{B_s\}_{s \in \mathcal{T}(a)}$, where each $B_s = (T|_{\ell:s(\neg\ell)})^W_{g|_{\ell:s(\neg\ell)}}$ (with the abuse of notation $T = T|_{\ell:s(\neg\ell)}$ referring to a local tensor block, and $g = g|_{\ell:s(\neg\ell)}$ the corresponding projection of the global random Gaussian vector, when the context is clear), to obtain the global eigenvector, we extract one eigenvector from a block at a time, then eliminate inconsistent eigenvectors.

To do that, we first vary the eigenvalue bound $\lambda$ and enforce the same lower bound $\lambda$ for each block $B$. Suppose we fix a bound $\lambda$. For each coordinate $i$, we want to have $|(Bx)_i| \geq \lambda |x_i|$, where $x$'s are the variables of the program. When $\lambda$ is tight for some block $B$, it must be the top eigenvalue for $B$ and the projection of $x$ on the coordinates in the block must be an eigenvector of $B$.

However, this requires $|x_i|$ to be correct, not just some upper bound on $|x_i|$. Therefore, we use the Rayleigh quotient definition of the eigenvalue, written as a semidefinite program (SDP) as follows:

$$\max \lambda \; s.t. \; \langle U_s, B_s \rangle \geq \lambda, U_s \in \Lambda, \text{ and } \langle U_s, U_s \rangle = 1 \text{ for all } s \in \mathcal{T}(a) \tag{7}$$

where $\Lambda$ is the set of all positive semidefinite matrices. We can factorize $U = V^\top D V$ for orthogonal matrices $V$ where $D$ is diagonal and $\langle U, U \rangle = 1$ enforces the diagonal to have norm 1. Then $\langle U, BU \rangle$ is maximized when $V$ contains a top eigenvector $v$ of $B$ and $D$ places all of the mass on the top eigenvector. We can also view this as having some common big matrix variable $U$ with lots of projections $P_s$ selecting out the blocks of $U$, and the constraints $\langle P_s U P_s^\top, \mathbf{B} P_s U P_s^\top \rangle \geq \lambda$, where $P_s$ selects the correct minor of $U$ for a big block $\mathbf{B}$ that contains all the small blocks $B_s$.

To find the correct $\lambda$, we will keep increasing $\lambda$ geometrically until the program is infeasible, i.e., $\lambda$ is too large. Then we use binary search on this range to find $\lambda$ that makes one block tight.

After finding the tight block and its eigenvector $e$, un-whiten it by applying $W^{-1}$ and record this top eigenvector as a fragment of the top global eigenvector we want to retrieve. Then look through the pre-computed decomposed local vectors and find the vectors $\{\hat{e}_{i'}\}$ that are inconsistent with the current tight vector. Then we subtract the weighted rank-1 matrix of each inconsistent $\hat{e}_{i'}$ (contracted with the same Gaussian vector) from its corresponding block $B'$:

$$B' - w_{i'} \langle \hat{e}_{i'}, g \rangle \hat{e}_{i'}^{\otimes 2} = \sum_{i=1}^r w_i \langle \hat{e}_i, g \rangle \hat{e}_i^{\otimes 2} - w_{i'} \langle \hat{e}_{i'}, g \rangle \hat{e}_{i'}^{\otimes 2} = \sum_{\forall i \in [r], i \neq i'} w_i \langle \hat{e}_i, g \rangle \hat{e}_i^{\otimes 2} \tag{8}$$

We then drop the constraints corresponding to the tight $B$, and we will continue the same procedure for the rest of the blocks. When all blocks have been tightened and dropped, we will collect all the un-whitened vector pieces together as one global vector $e$, and subtract $\lambda_m e_i^{\otimes 2}$ from each block, where $\lambda_m = \min_i \lambda_i$. We will collect $p = \lambda_m / \langle g, e_{i_m} \rangle$ as the probability for this global effect vector $e$, where $i_m = \operatorname{argmin}_i \lambda_i$. Then we will go back to the loop of tightening all the blocks again. We will continue doing this until all the blocks are reduced to zero.

To show the correctness of this algorithm, we need to show that the algorithm finds a consistent decomposition and terminates in polynomial time:

**Lemma 2.** *After each iteration of the inner loop, some $\hat{e}_{i'}$ remains consistent with $e_i$ in each block.*

*Proof.* First, we note that if a post-whitening 0-1 effect vector $v$ is not orthogonal to another post-whitening 0-1 effect vector $v'$, then $v$ and $v'$ are consistent: Contrapositively, if $e_i$ and $e_j$ are distinct effect vectors in some block, we have chosen $W$ so that $We_i$ and $We_j$ are orthogonal.

As in the analysis of Jennrich's algorithm, the eigenvalues of the contracted blocks are distinct almost surely. Recall that positive-semidefinite matrices are a conic combination of rank-1 matrices. Thus, for a fixed $\lambda$ that is tight for a block, the solution to the SDP equation 7 is a rank-1 matrix on that block, which is the tensor square of some vector $v = We_i$. For each other block, since the norm of the $U_s$ matrix is 1, the $U_s$ are convex combinations of tensor squares of vectors, where these $U_s$ are consistent with $v^{\otimes 2}$ on any common coordinates. In particular, as above, the whitening of the corresponding projection of $e_j \neq e_i$ onto the block gives a vector orthogonal to $We_i$. If the tensor $B_s$ for $s$ only had support on components inconsistent with $U_s$, we would have had $\langle U_s, B \rangle = 0$, but $\langle U_s, B \rangle \geq \lambda > 0$. Therefore, for each block $s$, $U_s$ must have positive weight on a rank-1 component that is the whitening of an effect vector consistent with $e_i$. $\square$

**Lemma 3.** *The outer loop terminates after at most $r|\mathcal{T}(a)|$ iterations.*

*Proof.* At the end of each outer iteration, we subtract $\lambda_m x_{i_m}^{\otimes 2}$ from each block. We notice that $\lambda_m$ is the eigenvalue and $x_{i_m}$ is the corresponding eigenvector of $A_{i_m}$. Therefore, the eigenvector is eliminated from $A_{i_m}$'s spectral decomposition. At the end of each iteration of the outer loop, we eliminate at least one of the (at most) $r$ eigenvectors of at least one of the (at most) $|\mathcal{T}(a)|$ blocks. Therefore, within $r|\mathcal{T}(a)|$ iterations, all blocks will be reduced to 0. $\square$

**Lemma 4.** *The effect probabilities are consistent with each block for a state appearing in $\mathcal{T}(a)$.*

*Proof.* Due to the uniqueness of the tensor decomposition, each weight $w_i$ we compute is the same as the true value in the latent tensor decomposition $\sum_{i=1}^{r} w_i e_i^{\otimes d}$. Since the weight $w_i$ is the probability (see Eq. 5), the local distribution is consistent. $\square$

## 5 SAFETY AND COMPLETENESS

We now state the theoretical guarantees for Stochastic SAM. Due to space constraints, we discuss their proofs in the appendix.

**Theorem 1** (Safety). *The probability any plan of length at most $L'$ succeeds in Stochastic SAM action model is at most $(1 + \epsilon)$ times greater than under the true model $M^*$. In particular, all actions that are applicable in a plan under the Stochastic SAM model are applicable to $M^*$.*

Note that it's immediate that the policy is applicable, since the precondition we produce is only stronger than the original: If $\ell \in pre(a)$ for any action $a$, $\ell$ is also a precondition for $a$ in the learned action model.

**Theorem 2** (Approximate Completeness). *Fix a planner, and suppose that for the distribution $\mathcal{D}$ over problems in a domain $D$, the planner produces a policy that solves the problem with probability $p$ and runs for $L$ steps in expectation, the draw from $\mathcal{D}$, and the planner itself. Given $m \geq poly(|A|, |F|^{O(\log r)}, L, 1/\epsilon, 1/\delta)$ trajectories independently drawn from the planner on problems from $\mathcal{D}$, with probability $1 - \delta$, the action model we learn satisfies the following: when a problem $\Pi$ is sampled from $\mathcal{D}$ and we execute a policy of length at most $L/\epsilon$ that maximizes the probability of solving $\Pi$ in the Stochastic SAM model with $L' = L/\epsilon$, $\Pi$ is solved with probability at least $p - O(\epsilon)$ (over both the draw of $\Pi$ and execution in $M^*$).*

The proofs closely follow Juba & Stern (2022), as we discuss in the appendix. The only difference between the proofs of these theorems and Juba & Stern (2022) is that we change the dependence on the number of fluents $|F|$ to the dependence on the number of effects $|F|^{O(\log r)}$.

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

APPENDIX

## A  LINEAR INDEPENDENCE AMONG TENSOR POWERS OF BOOLEAN VECTORS

We begin by identifying the smallest linear dependences among each tensor power of Boolean vectors.

**Theorem 3.** *For $k \in \mathbb{N}$, let $S \subseteq \{0,1\}^{k+1}$ of size $|S| < 2^{k+1} - 1$ be given. Then the set $S^{\otimes k} = \{\vec{x}^{\otimes k} : \vec{x} \in S\}$ is linearly independent; moreover, the only linear dependences for sets of size $2^{k+1} - 1$ are of the form $\sum_{\vec{x} \neq \vec{0}} \alpha(-1)^{w(\vec{x})} \vec{x}^{\otimes k}$ for $\alpha \neq 0$ where $w(\vec{x})$ denotes the Hamming weight of $\vec{x}$, $\sum_i x_i$.*

*Proof.* We show this by induction on $k$. For $k = 1$, $|S| \leq 2$, and indeed for any two distinct $\vec{x}, \vec{y} \in \{0,1\}^2$, wlog we have in some coordinate $i$ $x_i = 1$ and $y_i = 0$. Then in any linear combination in which $\vec{x}$ has nonzero weight $\alpha$, the $i$th coordinate is $\alpha \cdot 1 \neq 0$, so $S^{\otimes 1} = S$ is indeed linearly independent. For $|S| = 3$, the vector $(0,0)$ would yield a linear dependence among two vectors which we see is impossible. Thus the set must be $\{(1,1),(1,0),(0,1)\}$. Suppose that $(1,1)$ has weight $\alpha \neq 0$. Then $(1,0)$ and $(0,1)$ must have weight $-\alpha$ to yield a linear dependence.

Given the claim holds for $k - 1$, we show it for $k$. Consider any set $S$ for which for some coordinate $i^*$, more than $2^k - 1$ of the members $\vec{x} \in S$ have $x_{i^*} = 0$. Consider any linear combination of $S^{\otimes k}$, $\sum_{\vec{x} \in S} \alpha_x \vec{x}^{\otimes k}$. Then in the slice for $i^*$ of $S^{\otimes k}$, we would have the linear combination $\sum_{\vec{x} \in S : x_{i^*} = 1} \alpha_x \vec{x}_{-i^*}^{\otimes k-1}$; since $|\{\vec{x} \in S : x_{i^*} = 1\}| \leq 2^k - 2$ by hypothesis, the induction hypothesis yields that these are linearly independent and the linear combination must be nonzero.

Now suppose some coordinate $i^*$ has $2^k - 1$ 1s. Supposing that each of the slices have a linear dependence, we see that the coefficient of $\vec{x}_{-i^*}^{\otimes k-1}$ in the linear dependence must be $\alpha(-1)^{w(\vec{x}_{-i^*})}$ by induction hypothesis. But then, we see that in the $(i^*)^k$ entry of the tensor, we must obtain $\sum_{w=1}^{k} \alpha(-1)^w \binom{k}{w} = \alpha((1-1)^k - 1) \neq 0$. Thus, any linear dependence must be for a set of vectors that has $2^k$ 1s in each coordinate.

But now we observe that we cannot have sufficiently high weight in all coordinates to obtain a linear dependence: indeed, to have at least $2^k$ 1s in all $k + 1$ coordinates, we would have total weight at least $(k+1)2^k$. But, in dimension $k + 1$, observe that the greatest weight of $2^{k+1} - 2$ distinct vectors is obtained by omitting the vector $\vec{0}$ and some weight-1 vector, which has total weight at most $\sum_{w=1}^{k+1} w \cdot \binom{k+1}{w} - 1 = (k+1)2^k - 1$ (indeed, recall, $\sum_{w=1}^{k+1} x^w \binom{k+1}{w} = (1+x)^{k+1} - 1$, and taking derivatives, $\sum_{w=1}^{k+1} w x^{w-1} \binom{k+1}{w} = (k+1)(1+x)^k$). Thus there is no linear dependence among $2^{k+1} - 2$ vectors.

For $2^{k+1} - 1$ vectors, we again must use all vectors but $\vec{0}$. We note that in each coordinate $i$, we must thus use the standard basis vector $\vec{e}^{(i)}$, for which $\vec{e}_{-i}^{(i)} = \vec{0} \in \{0,1\}^k$. Thus, we see that the slices for each $i$th coordinate have exactly $2^k - 1$ nonzero vectors participating, and hence by induction hypothesis have weights $\alpha(-1)^{w(\vec{x}_{-i})-1}$. To obtain a linear dependence, now, we see that $\vec{e}^{(i)}$ must have weight $-\alpha = \alpha(-1)^{w(\vec{e}^{(i)})}$ so that $\alpha + \sum_{w=1}^{k} \alpha(-1)^{w+1} \binom{k}{w} = -\alpha - \alpha((1-1)^k - 1) = 0$, as claimed. $\square$

Now we observe that for Boolean tensors, increasing the dimension cannot yield a smaller linearly dependent set; this yields our final claim in this section, Lemma 1.

**Lemma (1).** *For all $k, n \in \mathbb{N}$ and $S \subseteq \{0,1\}^n$, if $|S| \leq 2^{k+1} - 2$, then $S^{\otimes k}$ is linearly independent.*

*Proof.* We proceed by induction on $n$. For $n < k + 1$, we observe that we can embed $S$ into $\{0,1\}^{k+1}$ by introducing $0$ coordinates to each $\vec{x}$ to obtain a set of vectors $S'$, where the tensors $\vec{x}^{\otimes k}$ for $\vec{x} \in S$ are obtained on the minors of $\vec{x'}^{\otimes k}$ for $\vec{x'} \in S'$. Then for all $n \leq k + 1$, the claim follows from Theorem 3.

Supposing now that we have established the claim for $n - 1 \geq k + 1$, we proceed to show it for $n$ as follows: Suppose we have a linear dependence in $S$ of size at most $2^{k+1} - 2$, and consider each of the coordinate-wise projections $S_{-i} = \{\vec{x}_{-i} : \vec{x} \in S\}$. Since $S_{-i}$ has dimension $n - 1$, we know by our induction hypothesis that $S_{-i}$ is linearly independent. Therefore, in our linear dependence, we must have that for each $\vec{x}' \in \{0, 1\}^{n-1}$ in the image of the projection, there must be more than one member of $S$ such that the total weight in the linear combination sums to 0. But, since $S$ is a set of Boolean vectors, there are exactly two vectors in $\{0, 1\}^n$ that map to each $\vec{x}' \in \{0, 1\}^{n-1}$; we thus find that if one is present in $S$, the other must be as well so that they may cancel. But now, we see that indeed, for any $\vec{x} \in S$, all of the vectors $\vec{y}$ of Hamming distance 1 from $\vec{x}$ are also in $S$. Indeed, since all of $\{0, 1\}^n$ can be reached by a series of Hamming distance 1 neighbors from any member, $S$ must contain all of $\{0, 1\}^n$. Then $|S| = 2^n \geq 2^{k+2} > 2^{k+1} - 2$, a contradiction. $\qquad\square$

# B    STOCHASTIC SAM IS PROBABLY SAFE AND APPROXIMATELY COMPLETE: PROOFS OF THEOREMS 1 AND 2

Our main result for safety and completeness are essentially the Theorem 4 and Theorem 5 in Juba & Stern (2022). It's important to point out that they assume the independence of the effects on individual fluents, while we do not. The proof closely follows Juba & Stern (2022). We will only describe the major changes here.

The most important change we make is on Theorem 2 of Juba & Stern (2022), i.e., the safety claim: For any $\delta > 0$, any action applicable to the learned action model $M$ is applicable to the true action model $M^*$. In additional, with probability $1 - \delta/(2|F|^d|A|)$, the learned effect probability $\hat{p}$ is close to the true $p^*$ in the sense that $(1 - \epsilon)\hat{p} \leq p^* \leq (1 + \epsilon)\hat{p}$. This guarantees the effect probabilities we learn are close to the true effect probabilities. The rest of the proof follows with minor modifications.

Let $x = Pr[\ell_1, \ldots, \ell_d | \neg\ell_1, \ldots, \neg\ell_d, a]$ be the true moment. Let $\hat{x} = \frac{\#_a(\ell_1, \ldots, \ell_d \in s')}{\#_a(\neg\ell_1, \ldots, \neg\ell_d \in s)}$ be the empirical estimate of $x$. We want to make sure $\hat{x}$ is close to $x$. Let's develop multiplicative error to derive the uncertainty interval of $x$. By using the Azuma-Hoeffding inequality in a similar fashion as in Theorem 2 of Juba & Stern (2022)(replace their single literal $\ell$ with tuple $\ell_1, \ldots, \ell_d$), we can have that for a tuple $\ell_1, \ldots, \ell_d$, its moment deviates from its empirical estimate by $\gamma$: $|x^* - \hat{x}| \leq \gamma$ w.h.p. $1 - \delta$, where $\gamma = \sqrt{\frac{\ln(2/\delta)}{2\#_a(\neg\ell_1, \ldots, \neg\ell_d \in s)}}$. To guarantee that all tuple estimates deviates by this $\gamma$, we use union bound and get that with probability $1 - |F|^d|A|\delta$, all empirical estimates deviate from their true moments by $\gamma = \sqrt{\frac{\ln(2/\delta)}{2\#_a(\neg\ell_1, \ldots, \neg\ell_d \in s)}}$, which is equivalent to say that with probability $1 - \delta$, all empirical estimates deviate from their true moments by $\gamma = \sqrt{\frac{\ln(2|F|^d|A|/\delta)}{2\#_a(\neg\ell_1, \ldots, \neg\ell_d \in s)}}$.

To develop multiplicative error, we do the following computation:

$$\hat{x} - \gamma \leq x^* \leq \hat{x} + \gamma$$

which implies that

$$(1 - \gamma/\hat{x})\hat{x} \leq x^* \leq (1 + \gamma/\hat{x})\hat{x}$$

Let's denote $\epsilon = \gamma/\hat{x}$, then we have

$$(1 - \epsilon)\hat{x} \leq x^* \leq (1 + \epsilon)\hat{x}$$

We want $\epsilon = \gamma/\hat{x}$ to be small. To do that, let's note

$$\epsilon = \gamma/\hat{x} = \sqrt{\frac{\ln(2/\delta)}{2\#_a(\neg\ell_1, \ldots, \neg\ell_d \in s)} \cdot \frac{\#_a(\neg\ell_1, \ldots, \neg\ell_d \in s)}{\#_a(\ell_1, \ldots, \ell_d \in s')}} \tag{9}$$

$$= \sqrt{\frac{\ln(2/\delta)}{2\#_a(\ell_1, \ldots, \ell_d \in s')}} \cdot \sqrt{\frac{\#_a(\neg\ell_1, \ldots, \neg\ell_d \in s)}{\#_a(\ell_1, \ldots, \ell_d \in s')}} \tag{10}$$

$$\leq \sqrt{\frac{\ln(2/\delta)}{2\#_a(\ell_1, \ldots, \ell_d \in s')}} \cdot \sqrt{1/\mu_{min}} \tag{11}$$

Here $\mu_{min} = \frac{\epsilon'}{|A||F|^d}$ is a minimal rate of the tuple $\ell_1, \ldots, \ell_d$ changing from $\neg\ell_1, \ldots \neg\ell_d$. If it's below this threshold, then $Pr[\ell_1, \ldots, \ell_d | \neg\ell_1, \ldots, \neg\ell_d, a]$ is considered missing. We choose this lower bound so that by invoking union bound on the missing probabilities over action and tuples, we can argue that with high probability $1 - \epsilon'$ the trajectories drawn from the training set only invoke actions in states where there are no missing entries. Hence the action's preconditions are satisfied.

Therefore, to make $\epsilon$ is small, we just have to make sure $\#_a(\ell_1, \ldots, \ell_d \in s')$ is large enough. This can be achieved by sampling large amount of trajectories. Indeed, let's consider the independent random event that for a trajectory there is at least one tuple $\ell_1, \ldots, \ell_d \in s$. By invoking Chernoff bound in a similar fashion as in Lemma 2 of Juba & Stern (2022), we can get that large amount of trajectories implies large $\#_a(\ell_1, \ldots, \ell_d \in s')$, which implies multiplicative error bound $(1 \pm \epsilon)$ for some small $\epsilon > 0$.

Next, we develop the multiplicative error bound for the actual effect probabilities. First, let's notice that in our algorithm, the only operations we have are linear operations, which means we performed some linear operations on all the estimated moment values to obtain the estimated effect probabilities $\hat{p}$. Let's denote this linear operation as a matrix $A$, and develop the multiplicative deviation from the true effect probability $p^*$ to $\hat{p}$ (with abuse of notation, here $x$ means the vector of all moments):

$$(1 - \epsilon)A\hat{x} \leq Ax^* \leq (1 + \epsilon)A\hat{x}$$

which implies

$$(1 - \epsilon)\hat{p} \leq p^* \leq (1 + \epsilon)\hat{p}$$

Here $Ax^* = p^*$ because the uniqueness of tensor decomposition.

The last major change to the proof of Juba & Stern (2022) is the transition probability $Pr_{\pi,M}[T_i | T_{i-1}]$ in the derivation of the bounds for $\tilde{p}$ and $\hat{p}$ in their Lemma 4 and Theorem 4. Instead of using the product of individual $\ell$'s transition probabilities $Pr[\ell | \neg\ell, a]$ due to the independence assumption, we directly use the estimated effect probability $\hat{p}$, which is $(1 \pm \epsilon)p^*$. Therefore, we do not rely on the assumption of independence of the effects on individual fluents.

## C  AN EXAMPLE ILLUSTRATION OF THE EFFECTS-LEARNING ALGORITHM

We can consider a toy environment in which there are two Boolean fluents, top and left. Let's suppose there is an action updown that changes top to a uniform random value. Let's suppose that we order the four fluents as 1: top, 2: ¬top, 3: left, and 4: ¬left. The tensor has four $4 \times 4$ slices, but we will never encounter states with complementary fluents satisfied. These will be missing entries, which we'll mark as ?. (We do know they must be zero, but this helps illustrate the general problem.) The moments for the action updown are then as follows:

|        | top | ¬top | left | ¬left |
|--------|-----|------|------|-------|
| top    | 1/2 | ?    | 0    | 0     |
| ¬top   | ?   | ?    | ?    | ?     |
| left   | 0   | ?    | 0    | ?     |
| ¬left  | 0   | ?    | ?    | 0     |

Table 1: top slice

|        | top | ¬top | left | ¬left |
|--------|-----|------|------|-------|
| top    | ?   | ?    | ?    | ?     |
| ¬top   | ?   | 1/2  | 0    | 0     |
| left   | ?   | 0    | 0    | ?     |
| ¬left  | ?   | 0    | ?    | 0     |

Table 2: ¬top slice

The algorithm uses a random contraction, which we'll suppose here is performed by summing over distinct slices; note that the observed blocks only contain one literal per each fluent, and so the contractions we consider only add a combination of one of the first two slices to one of the second

|        | top | ¬top | left | ¬left |
|--------|-----|------|------|-------|
| top    | 0   | ?    | 0    | ?     |
| ¬top   | ?   | 0    | 0    | ?     |
| left   | 0   | 0    | 0    | ?     |
| ¬left  | ?   | ?    | ?    | ?     |

Table 3: left slice

|        | top | ¬top | left | ¬left |
|--------|-----|------|------|-------|
| top    | 0   | ?    | ?    | 0     |
| ¬top   | ?   | 0    | ?    | 0     |
| left   | ?   | ?    | ?    | ?     |
| ¬left  | 0   | 0    | ?    | 0     |

Table 4: ¬left slice

two slices, both of which are all 0 in the observed entries. So, the contracted observed blocks are rescaled minors of the first two slices.

Let's suppose that the algorithm first encounters the partial effect vector $[1\ ?\ 0\ ?]$, which has eigenvalue $1/2$. Since the effect of top to rules out the effect ¬top, we also fix the second component to 0. This eliminates the ¬top slice. Now, with the observed block given by top and ¬left, we find that the ¬left effect must have value 0 – actually, this block must also be tight with eigenvalue $1/2$ – so we obtain a partial effect vector $[1\ 0\ 0\ 0]$ with eigenvalue $1/2$, corresponding to the effect that sets top true with probability $1/2$. We subtract the tensor cube $[1\ 0\ 0\ 0]^{\otimes 3}$ weighted by $1/2$ from the tensor to obtain

|        | top | ¬top | left | ¬left |
|--------|-----|------|------|-------|
| top    | 0   | ?    | 0    | 0     |
| ¬top   | ?   | ?    | ?    | ?     |
| left   | 0   | ?    | 0    | ?     |
| ¬left  | 0   | ?    | ?    | 0     |

Table 5: top slice on second iteration

|        | top | ¬top | left | ¬left |
|--------|-----|------|------|-------|
| top    | ?   | ?    | ?    | ?     |
| ¬top   | ?   | 1/2  | 0    | 0     |
| left   | ?   | 0    | 0    | ?     |
| ¬left  | ?   | 0    | ?    | 0     |

Table 6: ¬top slice on second iteration

|        | top | ¬top | left | ¬left |
|--------|-----|------|------|-------|
| top    | 0   | ?    | 0    | ?     |
| ¬top   | ?   | 0    | 0    | ?     |
| left   | 0   | 0    | 0    | ?     |
| ¬left  | ?   | ?    | ?    | ?     |

Table 7: left slice on second iteration

|        | top | ¬top | left | ¬left |
|--------|-----|------|------|-------|
| top    | 0   | ?    | ?    | 0     |
| ¬top   | ?   | 0    | ?    | 0     |
| left   | ?   | ?    | ?    | ?     |
| ¬left  | 0   | 0    | ?    | 0     |

Table 8: ¬left slice on second iteration

Now the nontrivial constraints correspond to vectors in the ¬top slice, e.g., the minor of that slice with ¬top and left, which gives the vector [? 1 0 ?] with eigenvalue $1/2$. Again, the effect not top rules out the effect top, so this must be extended to [0 1 0 ?], and we eliminate the top slice, and the ¬top and ¬left minor, where the eigenvector assigns ¬left 0, is also tight with eigenvalue $1/2$. Thus, we obtain the vector [0 1 0 0] with eigenvalue $1/2$. Subtracting this off gives an all zero tensor and we are done. We see that we have thus obtained that the effects are top with probability $1/2$, and ¬top with probability $1/2$, which is correct.

