# OpenReview forum: "Stochastic Safe Action Model Learning"
_ICLR.cc/2024/Conference — Submitted to ICLR 2024_

### Official Review · Reviewer_Ksk7 · 2023-10-28

**Soundness:** 2 fair
**Presentation:** 1 poor
**Contribution:** 2 fair
**Rating:** 3
**Confidence:** 3

**Summary:**

This study investigates a very interesting topic and introduces an algorithm for learning stochastic planning models, specifically targeting domains with dynamics that are challenging to model manually. The proposed approach could efficiently learn from example trajectories, ensuring accurate and safe action modeling.

**Strengths:**

1. The research topic is interesting.
2. The theoretical analysis sounds good.

**Weaknesses:**

1. The manuscript requires substantial improvements in writing quality, with an emphasis on a more coherent logical structure.
2. The paper contains numerous grammatical errors, even within the abstract. For instance, on page 1, there's a repeated "the", "model" in the abstract should be "models", "at some point" should be "at some points", and "some other condition is satisfied" should be "some other conditions are satisfied".
3. Ensure that abbreviations are expanded upon their first use, for example, "PPDDL".
4. Once an abbreviation has been defined, it's redundant to reintroduce it; consider the case with "Stochastic Safe Action Model (SAM)".
5. The experimental section is lacking, making it challenging to evaluate the method's effectiveness.

**Questions:**

1. Could you clarify the meaning of "IPC probabilistic tracks"?
2. Is there a correlation between the level of stochasticity and model performance?
3. What is the relationship between effect probabilities and sample complexity?

---

> ### Author Response · Authors · 2023-11-21
>
> We appreciate the review. The following are the answers to the questions.
>
> IPC probabilistic tracks is a set of benchmark datasets for planning, including parking, satellite, and transport planning. In these real world planning problems, the number of effects for each action is less than 5. The number of effects is the tensor rank in our formulation. Constantly small rank is a key requirement in our algorithm.
>
> The second question is not very clear to us. What do you mean by “the level of stochasticity”?
>
> The sample complexity does not depend on true effect probabilities values, but the confidence threshold and success rate. If the number of samples is large enough then the computed effect probabilities will be closer to the true probabilities.

---

### Official Review · Reviewer_cva7 · 2023-10-29

**Soundness:** 3 good
**Presentation:** 2 fair
**Contribution:** 2 fair
**Rating:** 3
**Confidence:** 3

**Summary:**

The focus of the paper is on model learning in stochastic PPDDL. Here, the overarching goal is to learn a model of the domain from trajectories. The model here specifically refers to a set of preconditions and effects of taking a particular action. The trajectories are executed with a set of policies in a domain with discrete states. Each state is characterized by a set of boolean fluents. The goal of the paper is to learn a stochastic model where the probability of each effect is extracted from the data. Previous work in this setting provides safety and approximate completeness guarantees by assuming that each effect’s action on each fluent is an independent random variable. This assumption eases the analysis. In contrast, this paper attacks a more challenging case using tools from tensor algebra. By performing a low-rank decomposition of the transition probability tensor using the method of moments, the authors are able to extract a model that is shown to satisfy safety and approximate-completeness criteria.

**Strengths:**

1) The contribution is novel, clear and significant. The idea of using tensor decompositions for PDDL has not been explored.

2) The method is theoretically sound.

**Weaknesses:**

1) The presentation and clarity needs significant improvement. As a standalone contribution, the paper should be more rigorous in terms of presentation and lacks a diligent writing style. A more scrupulous approach to explaining all the math will help presenting the paper (with the appendix).

2) It would be nice if half a page of the paper is delegated to demonstration of the method on one dataset.

3) More preliminaries and related work on the method of moments algorithm applicable to tensors is encouraged. The related work section only attributes around five papers.

**Questions:**

1) I have some questions surrounding Lemma 1. I believe $|S|$ denotes the number of distinct elements in $S$. Is there any reason why the elements of $V$ would not be distinct? Are they necessary to be all distinct? Is all that is sufficient is that $rank(V)=r$ where $r$ satisfies Lemma 1? In that case, $rank(V)+2 rank(V^{\otimes k}) \geq 3r$? This part is unclear to the reader.

2) More illustrations similar to section 3.2 equations (2) and (3) will help improve clarity.

3) Section 4.1 is not explained properly and there are some cyclical arguments. Given that these are mainly a variation of Jennrich’s algorithm, a preliminaries section can help ease the exposition.

4) There is no explanation of what is a “generic” tensor? Is the qualification in Kruskal’s theorem?

---

> ### Author Response · Authors · 2023-11-21
>
> We appreciate the review. Here are some explanation of the details of our tensor decomposition based method.
>
> The tensor decomposition will not return a non-distinct set of vectors V, because if there are two identical vectors, their weights will simply merge together in the decomposition, as we are seeking the minimum number of decomposed vectors.
>
> $Rank(V)=r$ would indeed be a sufficient condition but it’s generally not true. We do not make such assumptions but instead raise the moment to a degree that is large enough to ensure such rank lower bound.
>
> Could you please point out some of the cyclical arguments in Sec.4.1. so that we can further clarify?
>
> Generic tensor is the set of all tensors with the exclusion of a measure zero set. It is a term that has been used by the literature for the convenience of mathematical proof. We also found this term ambiguous since binary valued tensors can possibly fall into this set of measure zero. Therefore, we seek to provide a sufficient guarantee for the uniqueness of tensor decomposition in Sec.2.2, and Sec. 2.3.

---

> > ### Comment · Reviewer_cva7 · 2023-11-23
> >
> > Overall, after reading all the reviews and author comments, i believe there is progression over the discussion phase. However, I would like to retain my scores as the initial submission was unnecessarily cryptic and there are significant changes needed to improve the score further,

---

### Official Review · Reviewer_YR2D · 2023-10-31

**Soundness:** 2 fair
**Presentation:** 1 poor
**Contribution:** 2 fair
**Rating:** 3
**Confidence:** 4

**Summary:**

The paper presents an approach based on tensor decomposition for learning stochastic action models for symbolic planning. The problem is really relevant and important, given the amount of work going on in different fields model learning like learning abstractions or learning symbolic models.

The paper theoretically shows that the learned model is safe (or conservative) in terms of the action only applicable in a state if and only if it is permissible in the true model (but given that they learn a conservative model from a set of only positive trajectories this is not surprising).

**Strengths:**

- The problem is relevant, important and unsolved.

- The approach is theoretically sound and strong.

**Weaknesses:**

As I mentioned, the problem is really interesting. However, the paper is equally inaccessible to a reader. The low novelty score given is because even though the paper may have novel contributions, these are not understandable for the reader.

- There are many unsubstantiated claims in the paper. Theorems and Lemmas in the paper have almost no explanations.  While I support having theoretical results in the paper, they should be complete. The readers should not be left reading some previous work to understand even the basic premise of the theoretical results of the paper (in this case [Juba and Stern, 2022]) as the paper  does not have proofs for theorems and lemmas (Theorem under 2.2, Lemma 1, Theorem 1, and Theorem 2) or defer proofs to previous work.

- The notations are non-intuitive. For, e.g., the preliminaries section is meant to be make the rest of the paper understandable. However, they have unproven lemmas and theorems as well as equations with undefined symbols (superscript cross d ). In Theorem under Sec 2.2, what are a_k,b_k and c_k?

- The paper attempts to solve a very intuitive problem with a very non-intuitive approach. The most intuitive thing  would have been to include a running example that makes it really easy for the readers to follow.

- The next big problem with the paper is a lack of empirical evaluation. Without an empirical evaluation, there is no practical explanation to if the approach is feasible for learning real world domain models. There are plenty of PPDDL domains available to learn.

- The paper presents a similar functional approach as [Juba and Stern 2022] with near similar theoretical guarantees. It is not clear from the paper what is the motivation behind a different approach without any significant improvements.

**Questions:**

Please refer to the weaknesses highlighted in the previous section.

The most important question is:

- Would it be possible to provide a running example in the **main paper** to help the reader understand the paper as it is currently extremely difficult to understand.
- Why was not empirical evaluation provided and would it be possible to provide empirical evaluation on standard PPDDL domains?

---

> ### Author Response · Authors · 2023-11-21
> **It's not clear if this problem is intuitive**
>
> We appreciate the interest in the symbolic learning aspect of our paper.
>
> However, we would like to point out that our solution to this problem is not very intuitive. Firstly, even if the learning data consist of only positive trajectories, the success rate can still go wrong if the learned effect distribution of action model is off. Besides, the sampled policy that we used to produce the trajectories in the training set does not always end up reaching the goal state: the agent can simply reach the maximum number of steps and stop (please see the end of Sec. 2.1).
>
> The theorem under Sec.2.2 and Lemma 1 are classical tensor decomposition results. Theorem 1,2 both require long proofs which are not the focus of this paper. As long as we obtain the learning guarantee of the action’s effect distribution, we can plug it into their proof without much change. The main focus in this paper is how to learn each action’s effects without the strong assumption of independence of the environment factors.
>
> The superscript cross d means doing outer product with itself d times. $a_k, b_k, c_k$ are column vectors of the matrix.
>
> Again, this paper presents a completely different learning approach than Juba & Stern 2022. We are solving a problem with a more general setting, where the probability of the change of each environmental factor given an action is not necessarily independent of those of other factors (please also see our answer to reviewer bFXN).

---

> > ### Comment · Reviewer_YR2D · 2023-11-22
> > **Response**
> >
> > I thank the author for their response. They have tried to address some of the issues raised. However, most issues (even which they tried to resolve) are still unresolved.
> >
> > - The paper needs to be very clear about differences between Juba & Stern 2022 and the submitted work which it isn't. Even the response to the review is unclear.
> >
> > - The paper needs to be self sufficient, again, which it isn't. The presented theoretical paper heavily relies on Juba & Stern 2022 for most of the proofs which puts a lot of effort on the side of the readers. I do not think this is allows this paper to be published.
> >
> > - The authors claim that this paper solves the problem of learning a stochastic model in a  more general setting that Juba & Stern 2022, but imposes even stricter assumptions on the number of effect facts each action can have. They also fail to clearly outline this in the paper.
> >
> > - The authors in their response fails to acknowledge or comment on the need of a running example.
> >
> > Due to all this issues, I will like to keep my score as it is.

---

> ### Author Response · Authors · 2023-11-23
> **Further clarification**
>
> Thank you for the response. We would strongly recommend reading our answer to reviewer bFXN, as it clarifies the difference between our paper and Juba & Stern 2022, and more importantly, why their theoretical proof of safety and completeness is not our focus and we do not heavily rely on that. Regardless, we will simply repeat it as follows.
>
> Please note that the learning algorithm we propose has very little to do with the learning method proposed by Juba & Stern (2022). Their learning algorithm is simply computing the empirical estimates of $Pr( \ell’|a, \ell)$ for all the environment factor $\ell’$ and $\ell$ to reconstruct the transition probabilities $Pr(s’ | a, s)$, which is a product of $Pr( \ell’|a, \ell)$ probabilities, since they assume that the environment factors \ell are independent. This is not the case in our problem because we lifted the restriction of the factors being independent. If we simply compute the empirical estimates of all the transition probabilities $Pr(s’ | a, s)$, it will be infeasible since the number of states is exponentially large. Our tensor decomposition based learning method allows us to infer the set of effects of each action by only estimating a polynomial number of moments with bounded degree.
>
> The key that enables the use of Juba & Stern (2022)’s proof of safety and approximate completeness is the learning guarantee of each action’s set of effects in Alg.1 in this more general stochastic action model, which is the focus of this paper. Juba & Stern (2022) did not need such guarantee for their proof of safety or completeness, since their strong assumption of the independence of state factors allows them to simply learn the action’s effect on each factor independently.
>
> Because we do not assume such independence. Our setting is more general. The assumption of small number of effects for each action is justified in all the realistic benchmark datasets for planning in IPC probabilistic tracks, including parking, satellite, and transport planning, where the number of effects for each action is less than 5.
>
> Thank you for the response! As for running example, please see illustrations in section 3.2 equations (2) and (3), and a toy example that we included in Appendix C of the revision.

---

### Official Review · Reviewer_bFXN · 2023-11-02

**Soundness:** 3 good
**Presentation:** 1 poor
**Contribution:** 2 fair
**Rating:** 3
**Confidence:** 2

**Summary:**

In this paper, the authors introduce the problem of learning an action model in a stochastic environment of a PPDDL-type planning problem. Unlike the more standard MDP formulations of RL, here the state formulation consists of a set of 'fluents' which take boolean values, and the action model describes which 'effects' can follow after taking certain actions in given 'preconditions'. Compared to previous research in learning action models, in their formulation, the stochasticity of the effects that follow certain actions can be more general. The authors then show that, under these assumptions of the stochasticity, following closely the methodology of Juba & Stern (2022), they can learn an action model using tensor decomposition. They analyze the method and show that it can be used to achieve a particular notion of 'safety' and 'approximate completeness'.

**Strengths:**

* The authors come up with an algorithm to learn the action model and they can then guarantee "safeness" and the "approximate completeness" of the approach.

**Weaknesses:**

* The paper is unnecessarily dense at times, please consider the use of examples and captions to illustrate the main ideas, especially to new audiences.

* No experiments were performed to show the benefits of the introduced algorithm.

* It is not clear at times what the contribution is compared to Juba & Stern 2022 paper. It seems that all the proof techniques rely on that previous paper. In particular note the last sentence of the paper: "The only difference
between the proofs of these theorems and Juba & Stern (2022) is that we change the dependence on
the number of fluents |F | to the dependence on the number of effects |F |O(log r)."

* It is not clear if the stochastic model considered reflects real-world problems accurately. In particular it would be nice for the authors to give an example of a real-world problem that is captured by the particular stochastic model.

**Questions:**

* I'm not sure that ICLR is a good conference to submit this type of paper, it seems rather to belong to the more standard AI/planning-focused conferences.

* Is the Algorithm1 the authors' contribution, or is it also based on the Juba & Stern (2022) paper?

* It's not clear if the proposed algorithm would actually run on a computer. Have the authors tried to do so? Are there any complications?

* Minor comment: Two 'the's in the first sentence.

---

> ### Author Response · Authors · 2023-11-21
> **The focus of this paper is not repeating the proof of Juba & Stern (2022)**
>
> Thank you for pointing out the confusing part of this paper.
>
> When you say “methodology of Juba & Stern (2022)”, do you mean the learning method or the method to prove safety and approximate completeness?
>
> If you meant the learning method, please note that our assumptions of a more general stochasticity does not help us learn, and the learning algorithm we propose has very little to do with the learning method proposed by Juba & Stern (2022). Their learning algorithm is simply computing the empirical estimates of $Pr( \ell’|a, \ell)$ for all the environment factor $\ell’$ and $\ell$ to reconstruct the transition probabilities $Pr(s’ | a, s)$, which is a product of $Pr( \ell’|a, \ell)$ probabilities, since they assume that the environment factors \ell are independent. This is not the case in our problem because we lifted the restriction of the factors being independent. If we simply compute the empirical estimates of all the transition probabilities $Pr(s’ | a, s)$, it will be infeasible since the number of states is exponentially large. Our tensor decomposition based learning method allows us to infer the set of effects of each action by only estimating a polynomial number of moments with bounded degree.
>
> If you meant the latter, then please note that the key that enables the use of Juba & Stern (2022)’s proof of safety and approximate completeness is the learning guarantee of each action’s set of effects in Alg.1 in this more general stochastic action model, which is the focus of this paper. Juba & Stern (2022) did not need such guarantee for their proof of safety or completeness, since their strong assumption of the independence of state factors allows them to simply learn the action’s effect on each factor independently.
>
> Please note that PPDDL has an established action model that applies to the real world. Our model is strictly more general, which at least captures these real world situations and beyond.
>
> We agree that experiments will make this work stronger. However, the focus of this work is to derive a provable learning guarantee for the complex environment we consider, where it was not clear even a polynomial-time learning algorithm exists for this setting.

---

> > ### Comment · Reviewer_bFXN · 2023-11-22
> > **my rating remains the same**
> >
> > I thank the authors for their reply, in particular the sentence beginning with: "Their learning algorithm is simply computing the empirical estimates..." was quite clear and made me understand the difference in methodology between Juba & Stern (2022) and the current paper.
> >
> > I would have wished the paper to be equally clear but alas, it was quite difficult for me to make sense of the method and the proof. As the other reviewers also mentioned, a simple toy example together with some simulated experiments would greatly help. As of now, it is not suitable for publication in ICLR, which is a very interdisciplinary conference (as most ML conferences are) and requires very clear writing (such that people from many different backgrounds can understand the content). I keep my score as a result unchanged.

---

> > > ### Author Response · Authors · 2023-11-23
> > > **Toy example for illustration**
> > >
> > > Thank you for the response! As for running example, please see illustrations in section 3.2 equations (2) and (3), and a toy example that we included in Appendix C of the revision.

---

### Meta-Review · Area_Chair_AZuD · 2023-12-06

**Metareview:**

Summary of paper: This paper focuses on learning stochastic models (specifically pre-conditions and effects of taking actions) in Probabilistic Planning Domain Description Language (PPDDL) from trajectory data. The key idea is to use low-rank decomposition of the transition probability tensor. The paper’s main results are theoretical proofs that it can meet a specific definition of safety and completeness criteria.

Strengths: All reviewers agree that the problem studied is of importance. They also appreciated the ideas of using decomposition.

Weaknesses: All reviewers raised the same two key concerns: 1) the difficulty in understanding the content and disentangling what was prior work vs. the proposed contributions, 2) the lack of concrete examples of the proposed approach within the main text of the paper as well as empirical results substantiating that the proposed algorithm.

**Justification For Why Not Higher Score:**

After I carefully reviewed the discussion between the authors and reviewers, as well as the manuscript, I agree with the reviewer's recommendation for rejection at this time---the contribution needs to be more clearly articulated and the lack of concrete examples of the proposed approach raise questions about the feasibility of the method. With improvements to how this paper is written and positioned, as well as adding running concrete empirical examples, this paper could be strengthened for a future venue.

**Justification For Why Not Lower Score:**

N/A

---

### Decision · Program_Chairs · 2024-01-16

Reject